# The Effect of Chronotype on Risk-Taking Behavior: The Chain Mediation Role of Self-Control and Emotional Stability

**DOI:** 10.3390/ijerph192316068

**Published:** 2022-12-01

**Authors:** Qinfei Zhang, Xu’an Wang, Lvqing Miao, Lichun He, Huarong Wang

**Affiliations:** 1Department of Environmental and Health Psychology, Institute of Special Environmental Medicine, Nantong University, Nantong 226019, China; 2Xinglin College, Nantong University, Nantong 226014, China; 3Laboratory Animal Center of Nantong University, Nantong 226019, China

**Keywords:** chronotype, risk-taking behavior, self-control, emotional stability

## Abstract

Background: Although previous studies indicate that chronotype might be associated with risk-taking behavior, the specific mechanism has not been thoroughly discussed. This study aimed to fill this gap by exploring the mediating role of self-control and the chain mediating role of self-control and emotional stability between chronotype and risk-taking behavior. Methods: A total of 547 Chinese college students between 18 and 24 years old were selected to complete the Morningness–Eveningness Questionnaire (MEQ), Self-Control Scale (SCS), Eysenck’s Personality Questionnaire-neuroticism (EPQ-N), and Adolescent Risk-Taking Questionnaire: Risk Behavior Scale (ARQ-RB) to assess chronotype, risk-taking behavior, self-control, and emotional stability, respectively. Hayes’ PROCESS macro for SPSS was used to test the relationships among these variables. Results: Our result showed significant positive correlations among chronotype, self-control, emotional stability, and significant negative correlations between self-control, emotional stability, and risk-taking behavior. We also found that chronotype had a significant predictive effect on risk-taking behavior in the chain mediation model. Specifically, chronotype affected risk-taking behavior through two pathways: the separate mediating role of self-control and the serial mediation pathway of self-control → emotional stability. Conclusions: Our study provides direct evidence that chronotype is associated with risk-taking behavior. The results showed that the predictive function of chronotype was mediated by self-control and emotional stability. This study provides a new perspective on preventing and reducing risk-taking behavior.

## 1. Introduction

Risk-taking behavior is defined as either a socially unacceptable volitional behavior with probably undesirable consequences without preventive measures, such as speeding, drinking and driving, etc., or a relatively socially accepted behavior in which the danger is acknowledged, for instance, competitive sports, skydiving, etc. [1]. A considerable number of studies have underlined the negative outcome of risk-taking behavior among college students, such as reckless behavior, criminal activities, smoking, heavy drinking and drug use and abuse, which may have negative long-term consequences [2,3,4]. Therefore, research pertaining to factors influencing risk-taking behavior has theoretical and practical implications for reducing the occurrence of such behaviors and mitigating the associated consequences.

### 1.1. Chronotype and Risk-Taking Behavior

Several factors, including personality traits [5], sex differences [6], family structure, history of childhood trauma [7], contribute to risk-taking behavior. Among them, chronotype or morningness–eveningness preference is an important factor [8]. It is an individual behavioral manifestation of underlying circadian rhythmicity reflecting the time of the day at which individuals are “at their best” [9,10]. Chronotype is a spectrum ranging from extreme morning to extreme evening, with the remainder of the population falling in between these with intermediate typology (60–70%) [11].

A person with a morning preference who prefers morning activities, gets up easily and is more alert in the morning than in the evening, whereas a person with an evening preference prefers afternoon-evening activity, is more alert at night, and sleeps late into the morning [12]. Investigations in chronopsychology have revealed important differential results between morning-type and evening-type persons [13,14,15]. Previous findings suggest that morning-type traits are inversely related to greater risk-taking propensity [16], whereas evening-type individuals have significantly higher self-reported risk propensity [17]. Moreover, evening-types were significantly associated with higher impulsivity [13], which has a decisive role in the occurrence of risk-taking behavior. Thus, there may be significant differences in risk-taking behavior between morning and evening individuals. Although previous studies have examined the relationship between chronotype and risk-taking behavior, the effects of chronotype on risk-taking behavior has not been observed consistently in different domains and across studies. For example, although eveningness was previously reported to be associated with a greater propensity to take risks in financial, ethical, and recreational domains [18,19,20], the Morningness–Eveningness Questionnaire (MEQ) score found no association with scores obtained on the Balloon Analog Risk Task (BART), which is designed to measure risk-taking in terms of monetary gains [16]. The study adopted a general Chinese college student sample to investigate the relationship between chronotype and risk-taking behavior.

### 1.2. Self-Control as a Mediator

One potential underlying mechanism of the association between chronotype and risk-taking behavior might be self-control. Existing evidence highlighting the association between chronotype and self-control showed a significant and positive correlation between Morningness–Eveningness Questionnaire (MEQ) score and the Self-Control Scale score, indicating a relationship between evening-type and low self-control [21] and a relationship between morning-type and high self-control [22]. Another study reported similar regularity, whereby morning-type individuals have a higher level of self-control than evening-type individuals [23]. Differences in impulsivity between morning-type and evening-type might be attributed to a basic biological mechanism that determines an individual’s self-control [24]; indeed, other research has linked glucose to circadian rhythms [25] and glucose is the primary energy source that fuels self-control actions [26], and is thus considered to be associated with self-control. Circadian preferences may lead to variations in self-control performance [27]. In addition, self-control has been found to mediate the relationship between socioeconomic status and adolescents’ risk-taking behavior independently, whereby self-control is a negative predictor of the occurrence of risk-taking behavior [28]. Moreover, it has been suggested that improvements in self-control help to reduce risk-taking behavior, especially for teens with more parental conflicts or less parental support [29]. In addition, a high level of self-control closely related to the cognitive control system may significantly buffer the negative effect of an adverse social stimulus which activates the social–emotional system in male adolescents’ risk-taking [30]. However, a partial mediation role of self-control was only found between morning-type and risk-taking behavior in the financial domain in a previous study [20], not involving sociability, health, or ethics. Given these facts, we speculated that chronotype might affect risk-taking behavior by affecting self-control.

### 1.3. Emotional Stability as a Mediator

Emotional stability has been defined as the extent to which individuals can remain calm and stable under pressure, and has been described in the literature as an individual’s degree of self-confidence, tolerance of stress, optimism, and self-consciousness [31]. A significant correlation between self-control and emotional stability was observed [32] because self-control was shown to help develop a higher level of emotional stability, promoting personal competence and productivity [33], or preventing emotional dissonance [34]. Individuals with higher self-control could effectively regulate their emotion both subjectively (ratings of negative emotions by self-assessment) and neurologically (functional connectivity strength between the amygdala and prefrontal areas) [35,36]. Another study pointed out that emotional stability and risk-taking behavior were significantly and negatively correlated, specifically, in the participants’ self-report and risk behavior test (BART), where emotional stability moderated the effects of secondary psychopathy on risk preferences and risk-taking behavior [37]. In this study, we constructed a chain mediating effect model to test whether chronotype influenced risk-taking behavior through self-control and emotional stability.

Although previous studies have indicated that chronotype might be associated with risk-taking behavior, the specific mechanism of the relationship between chronotype and risk-taking behavior has not yet been thoroughly discussed. This present study was designed based on a previous study to reveal the simple mediating role and the chain mediating role of self-control and emotional stability between chronotype and risk-taking behavior. Specifically, we proposed the following hypotheses:

**Hypothesis** **1.**
*Chronotype would directly and positively affect self-control.*


**Hypothesis** **2.**
*Self-control would directly and positively affect emotional stability.*


**Hypothesis** **3.**
*Emotional stability would directly and negatively affect risk-taking behavior.*


**Hypothesis** **4.**
*Chronotype would indirectly and negatively affect risk-taking behavior through self-control and emotional stability.*


## 2. Materials and Methods

### 2.1. Participants

The study participants comprised 547 students (197 males [36.01%] and 350 females [63.99%]) from Nantong University (Nantong city, Jiangsu province, China). Their ages ranged from 18 to 24 years. Using a convenience sampling method, the data were sampled by administering an anonymous electronic questionnaire on the WenJuanXing public online platform (https://www.wjx.cn, accessed on 6 April 2021), during 3 March 2021 to 26 March 2021. All participants provided informed consent prior to the electronic questionnaire.

The researchers explained the research purpose and schedules to the participants and informed them that their participation was voluntary. Moreover, the participants were assured that all questionnaires would be kept confidential, and that all data would be used for scientific research purposes only. This study was approved by the Ethics Committee of Nantong University (2020-041).

### 2.2. Measure

#### 2.2.1. Chronotypes

Chronotypes were categorized with the Morningness–Eveningness Questionnaire (MEQ) [38]. The Chinese version of the MEQ was translated and tested by Zhang et al. [39], and its reliability and validity were confirmed to be as high as those of the original version. The Cronbach α in the current study was 0.749. The MEQ scores ranged from 16 to 86, with higher scores indicating greater morning-type preferences, and lower scores indicating greater evening-type preferences.

#### 2.2.2. Self-Control

The revised version of Self-Control Scale (SCS) for Chinese college students [40] was used to measure self-control. The scale consisted of 19 items, which were derived or revised from the SCS developed by Tangney et al. [41], and has acceptable internal consistency. This scale contains four subscales: 6-item Impulse Control (α = 0.801), 3-item Healthy Habits (α = 0.677), 4-item Resist Temptation (α = 0.626), 3-item Achievement and Task Performance (α = 0.574), 3- item Restrained Amusement (α = 0.490). The statements were assessed on a five-point Likert scale that ranged from 1 (completely false) to 5 (completely true). According to the composite score, a higher score corresponds to stronger self-control. The Cronbach α in our study was 0.876.

#### 2.2.3. Emotional Stability

The 24-item Eysenck’s Personality Questionnaire-neuroticism (EPQ-N) revised by Gong et al. was used to assess college students’ emotional stability [42]. The scale has been proven to have good reliability and validity in Chinese settings. The statement was assessed on a sum of the responses (agreement or disagreement) for each item. To facilitate data processing, a reverse scoring method was adopted. In this study, a higher score reflects more stable emotions. The Cronbach α in our study was 0.915.

#### 2.2.4. Risk-Taking Behavior

The Adolescent Risk-Taking Questionnaire, developed by Gullone et al. [43], consisted of two subscales: Risk Behavior Scale (ARQ-RB) and Risk Perception Scale (ARQ-RP). In this study, the Chinese version of ARQ-RB translated and revised by Zhang et al. was used to evaluate the risk-taking behavior of Chinese college students [44]. The scale contains 17 items and 4 factors: thrill-seeking, reckless, rebellious, and antisocial. Each item is rated on a five-point scale ranging from 1 (never true) to 5 (always true), with a higher score corresponding to a higher risk-taking propensity. In the current study, the Cronbach α was 0.749.

### 2.3. Statistics

To test correlations among variables, descriptive statistics and Pearson correlation analysis were inspected in SPSS 21.0 (SPSS, Inc., Chicago, IL, USA).

To examine the mediation effect of self-control and emotional stability, Model 6 of the PROCESS macro in SPSS21.0 was utilized [45]. We conducted bootstrapping with 5000 resamples to determine the mediation effect. If the bias-corrected bootstrap 95% confidence interval (CI) did not include zero, it indicated a significant mediation effect at the level of α = 0.05.

## 3. Results

### 3.1. Common Method Deviation Test

To avoid common methodological deviations, the Harman single-factor method was used for statistical control [46]. The results of statistical analysis showed that the characteristic value of three factors is greater than 1, and the first factor explained a variation of 37.63%, which was less than the critical value of 40%. Therefore, we can exclude the influence of common method deviation on the results of this study.

### 3.2. Descriptive Characteristics of the Sample and Correlation Analyses

This cross-sectional study included 547 adult college students comprising 197 males (36%) and 350 females (64%). The questionnaire survey involved four dimensions: chronotype, risk-taking behavior, self-control, and emotional stability. Descriptive statistics of the sample are reported in Table 1. The results show that among the 547 participants, the average chronotype scores ranged from 17 to 83, their total self-control scores ranged from 24 to 95, their emotional stability scores ranged from 0 to 24, and their risk-taking behavior scores ranged from 0 to 68.

Pearson’s correlations explored the relations between all the variables of interest. As indicated in Table 1, chronotype was positively and significantly correlated with self-control (r = 0.284, *p* < 0.01) and emotional stability (r = 0.170, *p* < 0.01), but not significantly correlated with risk-taking behavior (r = 0.034, *p* > 0.05). Moreover, both self-control (r = 0.261, *p* < 0.01) and emotional stability (r = 0.237, *p* < 0.01) were negatively and significantly correlated with risk-taking behavior. The correlation outcomes supported the relationship between the variables included in the hypothesized model.

### 3.3. Mediation Effect Test

Table 2 shows the results of the mediation analyses. Regarding the prediction of self-control, chronotype was found to have a significant and positive association with self-control (*p* < 0.001, Hypothesis 1). Concerning the prediction of emotional stability, self-control was observed to show a significant and positive association with emotional stability (*p* < 0.001, Hypothesis 2). For predicting risk-taking behavior, self-control and emotional stability were found to be significantly associated with risk-taking behavior (*p* < 0.001, Hypothesis 3).

Table 3 shows the chain mediating effect of self-control and emotional stability on the relationship between chronotype and risk-taking behavior. The chain mediating effect of self-control and emotional stability between chronotype and risk-taking behavior was significant (effect = −0.019, 95%CI = (−0.036, −0.006), Hypothesis 4).

Figure 1 shows the mediating effects of self-control and the sequential chain mediating effect of self-control and emotional stability on the association between chronotype and risk-taking behavior. The results showed that all paths in this model were significant (*p* < 0.01), except for the association between chronotype and emotional stability (B = 0.009, *p* > 0.05). However, the direction of the indirect effect (ab = −0.019) was opposite to the direct effect (c’ = 0.106), indicating that there is a “suppressing effect” between chronotype and risk-taking behavior. The results showed that the positive relationship between chronotype and risk-taking behavior was significant, with self-control partially mediating the relationship between chronotype and risk-taking behavior and a significant chain mediating effect of self-control and emotional stability.

## 4. Discussion

This study was designed to explore the antecedents of risk-taking behavior by assessing the relationship between chronotype, self-control, emotional stability, and risk-taking behavior among Chinese college students. Unexpectedly, we did not find a significant correlation between chronotype and risk-taking behavior, indicating no simple linear correlation between them. Unlike the traditional mediating effect test, when the relationship between an independent variable and dependent variable is not significant, the indirect effect, called the “suppressing effect”, might still exist, generalizing the intermediary effect [47,48]. Therefore, other variables should be considered in the relational mode of chronotype and risk-taking behavior. Our results suggested that self-control and emotional stability were positively and significantly correlated with chronotype, but negatively and significantly correlated with risk-taking behavior. These results provide preliminary support for the hypothesized mediating role of self-control and the chained mediating role of two-sequential mediators, self-control and emotional stability, in the relationship between chronotype and risk-taking behavior.

### 4.1. Theoretical Implications

#### 4.1.1. The Mediating Role of Self-Control

The indirect effect results confirmed our hypothesis, showing that self-control plays a simple mediating role between chronotype and risk-taking behavior. Specifically, morning-type people have a higher level of self-control, which might play a role in counteracting or reducing their curiosity about inappropriate or negative events to establish the complete behavior patterns that reduce the likelihood of risk-taking behavior, while those with evening-type have a lower level of self-control and harm avoidance [49] and a lower tendency for conscientiousness, which might negatively relate to harm-related behavior [50]. These results indicate that morning-type individuals might have more self-regulation and control over their behavior [35,50], leading to less risk-taking behavior. Comparatively, evening-type individual may have more difficulty controlling their thoughts and behavior and could be more likely influenced by emotions into impulsive behavior, showing lower harm avoidance [49], and resulting in risk-taking behavior.

#### 4.1.2. The Chain Mediating Role of Self-Control and Emotional Stability

The chain mediating role of self-control and emotional stability on the relationship between chronotype and risk-taking behavior was supported. Consistent with previous research, we found that self-control significantly and positively predicted emotional stability [32], which significantly predicted risk-taking behavior [51]. A better emotional regulation ability conferred by self-control enables individuals to adapt to external circumstances and social contexts [52]. Individuals with fewer mood fluctuations and greater mood improvements can maintain a high level of emotional stability [53]. It is suggested in the emotional generalization hypothesis that emotional stability is negatively correlated with the incidence of risk-taking behavior [54].

Considering the mediation variable, chronotype can positively predict risk-taking behavior, which was contrary to our expectations and previous findings. We speculated that this result could be because the college students are usually characterized by “getting up early and sleeping late”, which might have led to insufficient sleep, earlier daily routines, and worse sleep deprivation. In addition, an irregular sleeping schedule is also one of their characteristics; this observation is most likely due to conditions brought on by the COVID-19 pandemic. Students were likely to have a greater possibility to sleep at home during the COVID-19 pandemic, but their sleep schedules were still disturbed [55]. Several studies on students found that both irregular sleeping schedules and poor sleep were associated with risk-taking behavior [56,57], and that sleep deprivation is also one of the factors that triggering negative emotions and risk-taking behavior [16].

### 4.2. Practical Implications

The results of this present study might represent an important step toward implementing psychological intervention for reducing risk-taking behavior and associated consequences. Since self-control and emotional stability might play a significant role in explaining the relationship between chronotype and risk-taking behavior, current evidence can be used to guide college students in preventing or reducing risk-taking behavior and the associated consequences by improving their self-control and emotional regulation skills. Although there is no intention of labeling evening-type individuals with negative behaviors, the present literature tends to show a certain correlation between evening-type individuals being more prone to enter such a vicious cycle. To fundamentally solve this issue, more awareness is needed for evening-type individuals to inform them about such risks to develop healthy sleeping habits. Workers in the psychological field should also pay more attention to the mental health of evening-type persons and promote activities that could guide them to maintain more effective social stability.

### 4.3. Limitations and Future Research Directions

Although the present study advances our understanding of the relationship between chronotype and risk-taking behavior, some limitations should be addressed. First, we relied on cross-sectional data; thus, conclusions regarding the cross-sectional findings must be interpreted cautiously. Additionally, the data were cross-sectional and could not be used to test causality. Future research might implement prospective and longitudinal designs, for example, monitoring these variables and their interactions in the post-pandemic period [58]. Second, we only recruited college students from one university. Including participants from other universities and fields might have yielded more interesting findings among these variables. In the future, better-designed studies with larger sample sizes are needed to increase the level of evidence and confirm our hypotheses.

## 5. Conclusions

This study found a positive and significant association between chronotype, self-control, and emotional stability. Further, there was a negative and significant association between risk-taking behavior, self-control, and emotional stability. In addition, the relationship between chronotype and risk-taking behavior was partially and sequentially mediated by self-control and emotional stability.

## Figures and Tables

**Figure 1 ijerph-19-16068-f001:**
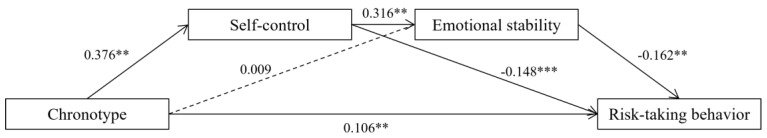
Model of the hypothesized mediator role of self-control and emotional stability in the relationship between morning–evening type and risk-taking behavior. All the path coefficients are standardized. ** *p* < 0.001, *** *p* < 0.001.

**Table 1 ijerph-19-16068-t001:** Descriptive statistics of the sample and correlations of all variables.

Variables	N	M	SD	Chrono-Type	Self- Control	Emotional Stability	Risk-Taking Behavior	Min	Max	Mean	SD	Skewness	Kurtosis
Gender													
Males	197												
Females	350												
Chronotype		46.640	8.882	1	0.284 **	0.170 **	0.034	17	83	46.640	8.882	−0.037	0.500
Self-control		60.830	11.753	0.284 **	1	0.562 **	−0.261 **	6	30	20.440	4.671	−0.065	−0.09
Emotional stability		14.360	6.600	0.170 *	0.562 **	1	−0.237 **	0	24	14.36	6.600	−0.236	−0.929
Risk-taking behavior		68.0	7.942	0.034	−0.261**	−0.237**	1	0	68	9.540	7.942	2.142	7.939

Note. * *p* < 0.05, ** *p* < 0.01.

**Table 2 ijerph-19-16068-t002:** The results from the mediation analyses.

Independent Variables	Predictor Variables			
	Risk-Taking Behavior	Self-Control	Emotional Stability	Risk-Taking Behavior
Chronotype	0.030	0.376 ***	0.009	0.106 **
Self-control			0.316 ***	−0.148 ***
Emotional stability				−0.162 **
R^2^	0.001	0.081	0.316	0.093
F	0.622	47.828	125.368	18.578

Note. ** *p* < 0.01, *** *p* < 0.001.

**Table 3 ijerph-19-16068-t003:** Standardized indirect effects and 95% CIs for the mediational model (*n* = 296).

Model Pathway	Effect	BootSE	BootLLCI	BootULCI
Chronotype→Self-control→Risk-taking behavior	−0.056	0.017	−0.092	−0.026
Chronotype→Emotional stability→Risk-taking behavior	−0.001	0.005	−0.013	0.009
Chronotype→Self-control→Emotional stability→Risk-taking behavior	−0.019	0.008	−0.036	−0.006

## Data Availability

The data that supports the findings of this study are available from the corresponding author upon reasonable request.

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
