# Peer review of "The Effect of Chronotype on Risk-Taking Behavior: The Chain Mediation Role of Self-Control and Emotional Stability"

_ijerph, 2022, doi:10.3390/ijerph192316068_

Round 1

Reviewer 1 Report

The Effect of Chronotype on Risk-Taking Behavior: The Chain Mediation Role of Self-Control and Emotional Stability

This study explored the mediating role of self-control and the chain mediating role of self-control and emotional stability between chronotype and risk-taking behavior.

I believe that the authors presented an interesting topic; they were neat in supporting their work, in the choice of variables, data preparation, and methods.

General suggestions

I suggest the authors check some spelling errors, punctuation, and transitions in their text. For example, in the abstract: “…and the chain mediating role self-control and emotional stability…”, it’s necessary to add the word “of” before self-control.

Also, I recommend checking the clarity of some phrases or making the writing simple. For example, you can use the subject-verb-predicate structure in the sentences, simplify your phrases, and use one idea per paragraph. That could make your text better.

Abstract

The conclusions in the abstract could be different from the results report, I mean, you can argue the reasons because it was rejected the null hypothesis, trying to link with the state of the art in this topic.

Introduction

In line 72, the phrase: “Emotional stability (also called neuroticism)”, I think it’s an incorrect statement because, in theory, neuroticism is the opposite of emotional stability. Also, I consider “neuroticism” is a very criticized term according to the recent literature, for example: https://doi.org/10.1073/pnas.1919934117

In lines: 79-80, the authors expressed “In addition, individuals with higher self-control could do well in emotion regulation both subjectively and neurologically”, I suggest explaining subjectively and neurologically, what do they mean?

I suggest reviewing writing in this sentence: “Based on previous studies, we designed this study to reveal the mediating role of self-control and the chain mediating role self-control and emotional stability between chronotype and risk-taking behavior.” (Lines 85-87).

Discussion

I suggest reviewing writing: “Therefore, when evening-type college students are making decisions, they could be more likely to experience problems such as diminished self-control and impulse, which further influence it less likely to have good adaptation or emotional stability, and those morning-type students who have regular schedule can reduce their negative effects.”

The authors should clarify the term “cultivation work”, it’s probable that in the West, the meaning is different from what they mean.

I consider that in the sentence: “To sum up, advocating regular schedule can not only improve the quality of life of college students, keep physical and mental health, but also help maintain social stability and even reduce crime rates.” (Lines 242-244), it’s necessary to complete the term: “regular schedule” of what activity? Also, it is important to consider if they jumped to a conclusion that was not justified by data, or previous research.

In lines 256-257, review the writing of the sentence: “…for example, we need for monitoring these variables and their interaction in the post-pandemic period…”

I consider these phrases are not necessary: “The findings and their implications should be discussed in the broadest context possible. Future research directions may also be highlighted.”

Author Response

请参阅附件。

Reviewer 2 Report

This stusy examined the association between Chronotype and Risk‐Taking Behavior , which is an interesting theme. However, some issues shodule be addressed before further processing:

(1) Why did you focus on  Chronotype, and its relation with Risk‐Taking Behavior shoudl be further examined;

(2) "the partial mediation role of self‐control was only found between morning‐type and risk‐taking behaviors in the financial domain in previous study the partial mediation role of self‐control was only found between morning‐type and risk‐taking behaviors in the financial domain in previous study ", what are the differences and contributions of this study toexamine the same issue? only this stiudy was conduceted in adolescents?

(3)I wonder wtheter the mediating mechanism is rational, though it has been Validated by the data. Rrelevant theoretical and emperical evidences is insufficient. How self‐control is associated with Chronotype is not clear.

(4) regarding the  Emotional stability, it's a typical personality trait, measured by Personality Questionnaire‐neuroticism in this study), adopting it as a mediating role is not  convincing. Maybe, moderating role is nore suitable.

(5)for the results, for the M, average value is more encouraged for the main variables.

(6) The discussion should be expanded, and the significance and contribution of this study shodul also be discussed.

Author Response

请参阅附件。

Reviewer 3 Report

The study has 3 hypotheses but the largest part was to investigate the mediating role of selfcontrol and emotional stability on the relationship between chronotype and risk-taking behavior.  To the best of my knowledge, there is no study that explored the relationship between chronotype and risktaking behavior by mediating the role of self-control and emotional stability.  Previous studies investigated the impact of other parameters on risk-taking behavior such as personality characteristics, differences in sex and gender, family culture, and trauma including childhood and adulthood. The studied parameter in the presented study “chronotype” is another variable that is taken into account because it relates the time of day to the best/worst behavior of individuals (which could be linked to risk-taking behaviors). The discussed methodology is appropriate (a simple mediation analysis and/or some correlation) for the hypothesis of the study. References are relevant. The tables are simple and clear.

Comment1: The authors mentioned that “Among them, chronotype is an important factor [8]” please explain clearly why/how chronotype is important.

Comment2: The authors mentioned that “The Chinese version of the MEQ was translated and tested by Zhang et al. [30], and its reliability and validity were confirmed to be as high as those of the original Version” please add a reference showing the validation is confirmed.

Reviewer 4 Report

The authors develop a conceptual framework that delineates the mechanisms by which chronotype has an effect on risk-taking behavior by identifying the chain mediating effect of self-control and emotional stability. However, I dont think this article can make a contribution in its current form, and I offer some suggestions to improve it.

1. The term of chronotype should be stated more clearly, morning-type or evening-type? Also, the positive or negative directions of the indirect effects should be specified.

2. The theoretical argumentation of this article is not compelling and must be greatly strengthened, especially the effects of chronotype on individuals ability of self-control and personality of emotional stability, and the influence of self-control on emotional stability.

3. The Hypothesis 1 is rather confusing. The authors state that morning-type traits are inversely related to greater risk-taking propensity, and higher scores on Morningness-Eveningness Questionnaire in the current study indicated greater morning-type preference. What is the Hypothesis 1: Chronotype can positively predict risk‐taking behavior based on?

4. Also, in the result section, the author claimed that Hypothesis 1 was supported. According to this results, morning‐type individual are inclined to engage in risk-taking behaviors and have a higher level of self‐control ability. This logic is confused.

5. Based on existing evidence, it is more reasonable to expect there will be a negative relationship between morning chronotype and risk-taking behavior, and a negative indirect effect through self-control and emotional stability. The authors found a positive direct association between chronotype and risk-taking , and  two negative  indirect pathways , in other words they revealed an unexpected suppressing effect, which ran counter to common sense and should be well-explained in the discussion section.

6. My another concern is that individual difference in chronotype has a strong  biological foundation. It is possible that the psychological individuals differences (e.g., personality and behavior pattern) associated with chronotype is attributed to the potential biological mechanism rather than chronotype itself.

Author Response

请参阅附件。

Round 2

Reviewer 2 Report

The quality of the manuscript has been imporved, there are still some issues needed to be addressed:

(1) subtitles are suggested in the introduction and discussion part to clearly present the information;

(2) the theoretical evidences shoudl be strengthened, especially for the hypohesized model;

(3) though you said that emotional stability (as a personality) could be adopted as a mediating role., the evidence is weak. Could you test wheter it could act as a moderating role in the model?

(4) in the figure, theinsignificant pathway should be presented in with dotted line;

Reviewer 4 Report

In the reversed version, the research hypotheses and results are  presented more clearly, also the authors have pointed out and explained the suppressing effect and fixed all errors. I really appreciate the effort the authors made for the improvement of this manuscript. I only have a few  minor suggestions  before  publication.

1. The authors should  explicly state the directions of the associations among variables in abstrat. For example,  "Results: Our result significant correlations among chronotype, self-control, emotional stability, and risk-taking behaviors".

2. Pleas check Table1. It shows that the correlation bewteen chronotype and risk-taking behaviors is 0.562 **. I think this is a mistake.

3. There are still some of grammatical mistakes, please improve the English.
